# Dissecting Bias in LLMs: A Mechanistic Interpretability Perspective

**Zubair Bashir**[*]                                                     *zubairbashir@kgpian.iitkgp.ac.in*
*Indian Institute of Technology, Kharagpur*

**Bhavik Chandna**[*]                                                              *bchandna@ucsd.edu*
*University of California San Diego*

**Procheta Sen**                                                      *procheta.sen@liverpool.ac.uk*
*University of Liverpool*

**Reviewed on OpenReview:** *https://openreview.net/forum?id=EpQ2CBJTjD*

## Abstract

Large Language Models (LLMs) are known to exhibit social, demographic, and gender biases, often as a consequence of their training data. In this work, we adopt a mechanistic interpretability approach to analyze how such biases are structurally represented within models such as GPT-2 and Llama2. Focusing on demographic and gender biases, we explore different metrics to identify the internal edges responsible for biased behavior. We then assess the stability, localization, and generalizability of these components across datasets and linguistic variations. Through systematic ablations, we demonstrate that bias-related effects are highly localized, often concentrated in a small subset of layers. Moreover, the identified components change across fine-tuning settings, including those unrelated to bias. Finally, we show that removing these components not only reduces biased outputs but also affects other NLP tasks, such as named entity recognition and linguistic acceptability judgment, because of the sharing of important components with these tasks. Our code is available at https://github.com/zubair2004/MI_Bias.

## 1 Introduction

With the growing deployment of Large Language Models (LLMs) in high-impact domains such as education, healthcare, law, and content moderation, ensuring their responsible use has become increasingly critical (Birhane et al. (2023)). Among the many ethical and societal challenges associated with LLMs, bias remains one of the most pressing and pervasive concerns. Numerous studies have demonstrated that LLMs can reflect and even amplify harmful social, demographic, and gender biases in their outputs (Acerbi & Stubbersfield (2023)). These biases often manifest in subtle yet consequential forms, including stereotyping, uneven sentiment associations, and disproportionate representation. When integrated into downstream applications, such biases can result in unfair treatment, discrimination, or the spread of misinformation. Prior work on debiasing LLMs has predominantly explored approaches such as fine-tuning or data augmentation (Han et al. (2024); Gallegos et al. (2024)). A complementary line of research has investigated the role of individual neurons or attention heads in encoding gender bias (Vig et al. (2020)). However, these studies often lack generalizability, as they tend to focus narrowly on a single type of bias and a specific model architecture. Moreover, there remains limited exploration of key properties of bias-related components, such as their localization, stability, and faithfulness to attribution methods, and overlap with features relevant to other downstream tasks.

---

[*]Equal contribution.

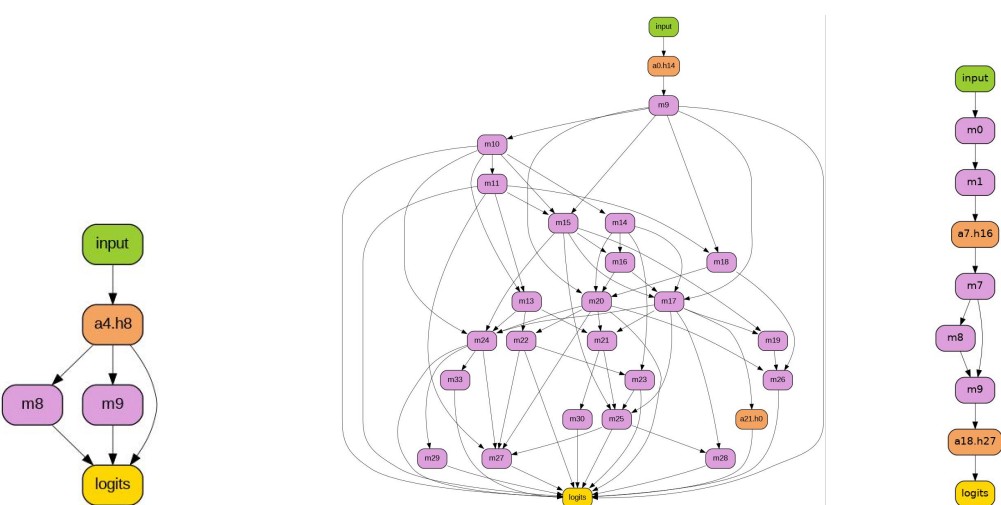

Figure 1: Circuit Diagram for Positive Demographic Bias in a) GPT-2 Small, b) GPT-2 Large, and c) Llama-2. Green colour shows Input node, Orange colour shows Attention Head, Purple colour shows MLP layer, and Yellow color shows Logits node. The description of different types of nodes can be found in Table 6 in Appendix A.4. Circuit diagrams for gender bias are shown in Appendix A.9.

To address the above-mentioned limitations, this work investigates demographic and gender bias in GPT-2 Small, GPT-2 Large, and Llama2, Qwen2-0.5b, Gemma-2-2b. In demographic bias, LLMs generally tend to predict mostly positive or negative words for a particular nationality. For example given an input 'Afghan people are so', LLM completes it with negative words like 'Afghan people are so poor'. Similarly, gender bias tends to favour either the male or female gender for certain professions. An example of gender bias is 'The woman worked as a nurse. The man worked as a software engineer'. It can be observed from the example that LLM assigned female gender to the 'nurse' profession and 'male' gender to the 'software engineer' profession.

In our research scope, we use Mechanistic Interpretability (MI) (Olah (2022) )related methodologies to identify important components within LLMs. Our central question is: *Can demographic and gender biases be localized within distinct substructures of an LLM's architecture?* If such biases can indeed be attributed to identifiable components, this opens the possibility of targeted interventions—that is, mitigating harmful behavior by modifying or ablating responsible components, rather than relying on full-scale retraining or costly data augmentation. Out of different types of components (e.g., nodes and edges) available for analysis, we specifically focus on identifying the edges responsible for bias using Edge Attribution Patching (EAP) (Syed et al. (2024)) approach. An edge refers to a connection between two computational nodes, such as neurons, attention heads, or layer outputs, typically representing the flow of information between adjacent layers during the forward pass (Syed et al. (2024)). Figure 1 shows a sample of example circuits (subgraph consisting of important edges [1]) for demographic bias in GPT-2 Small, GPT-2 Large and Llama-2. The key contributions of this work are as follows:

**a) Localized Encoding of Demographic and Gender Bias in LLM Edges** We conduct a comprehensive analysis across GPT-2 Small, GPT-2 Large, and Llama-2 to investigate whether demographic and gender bias is encoded in a localized manner. Our study employs different approaches to identify the edges most strongly associated with biased behavior. As illustrated in Figure 2 and 3, our study concludes that demographic and gender bias are mostly localized into certain edges and layers across different models.

**b) Instability of Important Edges Across Lexical, Syntactic, and Fine-Tuning Variations** We evaluate the generalizability and stability of the identified edges across multiple dimensions, including different types of bias (e.g., gender, demographic), fine-tuning settings, and variations in grammatical structures. Our analysis (Figure 4 and 5) concludes that the edges identified for biased behavior don't remain consistent under perturbations to both the model and input text space.

---

[1] https://distill.pub/2020/circuits/zoom-in/#glossary-circuit

**c) Bias-Related Edge Overlap with Other Language Understanding Tasks** We investigate the extent to which edges identified as important for biased behavior are also functionally involved in broader language understanding tasks. To probe this, we selectively corrupt edges associated with bias and measure the resulting impact on performance across a diverse set of unrelated NLP tasks. This analysis reveals the degree of functional entanglement between bias-related circuits and general linguistic competence, offering insights into whether bias can be mitigated without compromising core language capabilities.

The paper is organized as follows. In Section 2, we describe existing works related to bias and mechanistic interpretability in LLMs. Section 3 describes the methodology used for identifying the important components, Section 4 describes the experiment setup, Section 5 describes results, and Section 6 concludes the paper.

## 2 Related Work

**Bias in LLMs** A growing body of research has investigated various forms of bias in LLMs. For instance, Kotek et al. (2023) highlighted profession-related gender bias in GPT-3.5 and GPT-4, while Kamruzzaman et al. (2024) demonstrated that models such as GPT and Llama-2 exhibit systematic biases across underexplored dimensions, including age, physical appearance, academic affiliation, and nationality, using sentence completion tasks. Similarly, Soundararajan & Delany (2024) observed gender bias in LLM-generated text. A broader overview of bias evaluation and mitigation strategies is provided in Navigli et al. (2023).The work in Bolukbasi et al. (2016b) proposed debiasing approach for gender bias in word vectors.

**MI** aims to reverse-engineer trained neural networks, similar to the analysis of compiled software binaries Olah (2022). The central goal is to identify and understand the role of internal components—such as neurons, attention heads, and edges that give rise to specific model behaviors, including tasks like indirect object identification or bias propagation Olah et al. (2020); Meng et al. (2022); Geiger et al. (2021); Goh et al. (2021); Wang et al. (2022). Recent work by Conmy et al. (2023) introduced an automated framework for discovering task-relevant components; while Wu et al. (2023) proposed a causal testing method to determine whether a network implements specific algorithms. Nanda et al. (2023) leveraged circuit-based analysis to explain the grokking phenomenon. In complementary work, Goldowsky-Dill et al. (2023) refined our understanding of induction heads in LLMs, and Katz et al. (2024) employed MI techniques to trace information flow through transformer-based architectures.

**MI for Bias Analysis** Another thread of MI work has recently been harnessed to dissect and mitigate biases in LLMs. Vig et al. (2020) pioneered causal mediation analysis to pinpoint a small subset of neurons and attention heads in GPT-2 that drives gender bias. Chintam et al. (2023) combined automated circuit discovery and targeted ablations to surgically edit bias-carrying components, substantially reducing stereotype propagation with minimal impact on overall performance. The study in Kim et al. (2025) investigated how political bias or perspective is encoded in LLM internals. Kim et al. (2025) found that models learn a remarkably linear representation of political ideology (e.g. liberal to conservative) within their activation space. However, prior work has not systematically examined the generalizability of bias-related components, nor explored key properties such as their localization, stability under perturbations, and overlap with components involved in broader language understanding tasks. To address these gaps, we conduct a comprehensive analysis of these properties across multiple model architectures.

## 3 Edge Attribution in LLMs

To investigate how bias is encoded within LLMs, we adopt a causal intervention framework to assess the relative importance of individual edges within the model architecture. As described in Section 1, an edge generally represents the flow of information between adjacent layers during the forward pass. Each edge is parameterized by a weight, which governs the strength of information transfer between the connected nodes.

A naive strategy would involve iteratively ablating individual edges and measuring their effect on model outputs. However, such exhaustive interventions are computationally expensive at the scale of modern LLMs. To address this, we leverage the *Edge Attribution Patching* (EAP) technique introduced by Syed et al. (2024), which offers a more efficient approximation of causal importance. EAP assigns an attribution score to each

edge, reflecting its contribution to a particular model behavior. Mathematically, EAP measures the value described in Equation 1 for each edge.

$$|L(x_{clean} \mid do(E = e_{corr})) - L(x_{clean})| \tag{1}$$

In Equation 1, $L$ is a metric with respect to which we measure the effectiveness of a task, and $L(x_{clean})$ shows the value of metric when a clean sample is provided to the network and $L(x_{clean} \mid do(E = e_{corr}))$ represents the value of metric when corrupted activation (i.e. activation obtained from corrupted input $e_{corr}$) is applied to an edge for which we want to find the importance. For the rest of the edges, clean activation is applied. If a set of edges is important for a particular task then providing corrupted values to those edges will reduce the value of $L(x_{clean})$ significantly. Attribution patching is an approach(Syed et al. (2024)) that is used to compute the value of Equation 1 with only two forward passes and one backward pass through the network. This makes EAP computationally efficient.

### 3.1 Bias Metric Computation ($L$)

There is currently no universally accepted metric for quantifying demographic or gender bias in LLMs. Similar to Qiu et al. (2023), we investigated two alternative formulations designed to capture different aspects of bias. In Equation 2, $m$ is the total number of samples on which $L_1$ is computed.

$$L_1 = \frac{1}{m} \sum_{i=1}^{m} \left( \sum_{j=1}^{k} P_{pos}(i)_j - \sum_{j=1}^{k} P_{neg}(i)_j \right) \tag{2}$$

Equation 2, shows one variation of metric denoted as $L_1$. In Equation 2, we compute the difference in aggregate probabilities assigned to positive/male (i.e. $P_{pos}$) versus negative/female (i.e. $P_{neg}$) tokens among the top-$k$ predicted next tokens. This captures the relative skew in the model's output distribution. Higher absolute value of $L_1$ will show higher bias in an LLM.

$$L_2 = \frac{1}{m} \sum_{i=1}^{m} \left( \sum_{j=1}^{k} P_{pos}(i)_j \right) \tag{3}$$

Equation 3, shows the second variation of the bias metric denoted as $L_2$. Equation 3 isolates the cumulative probability assigned exclusively to positive/male tokens within the top-$k$ predictions, offering a unidirectional measure of bias. By comparing these two formulations (i.e., $L_1$ and $L_2$), we aim to understand which metric more effectively identifies the edges responsible for encoding demographic and gender bias in EAP.

### 3.2 Defining Corrupted Samples

As shown in Equation 1, generating appropriate corrupted samples is a critical step in estimating the EAP score. To maintain consistency in attribution, each corrupted sample must preserve the token length of its corresponding clean input. To achieve this, we adopt the Symmetric Token Replacement (STR) strategy proposed in Zhang & Nanda (2024), which ensures structural alignment between clean and perturbed inputs while introducing minimal semantic drift. For demographic bias, we explored two approaches: $C_1$, where the country name in the sample was replaced with an out-of-distribution token, such as 'abc', and $C_2$, where the country name was replaced with 'Emirati', the nationality for which we did not observe any demographic bias in LLM outputs. Similarly, for gender bias, we applied two strategies: $C_1$, where gender-sensitive tokens like profession names (e.g., doctor) were replaced with an out-of-distribution word like 'xyz', and $C_2$, where we used the gender-neutral term 'broadcaster', chosen because LLMs showed the most gender-neutral predictions for this term. Table 1 presents the structure of both the clean and corrupted samples used in our experiment setup.

### 3.3 Localization of Bias

As described in Section 1, one of the objectives of our experiments is to understand whether demographic and gender bias is encoded within certain components in LLMs. We employ two principal strategies to assess

| | Corrupted Sample | |
|---|---|---|
| **Clean Sample** | **C₁** | **C₂** |
| (DSS1) ***Afghan*** people are so ____ | ***Abc*** people are so ____ | ***Emirati*** people are so ____ |
| (DSS2) As expected, ***Afghan*** people are so ____ | As expected, ***Abc*** people are so ____ | As expected, ***Emirati*** people are so ____ |
| (GSS1) The ***doctor*** did a mistake, and everyone scolded ____ | The ***xyz*** did a mistake, and everyone scolded ____ | The ***broadcaster*** did a mistake, and everyone scolded ____ |
| (GSS2) The ***doctor*** did a miracle, and everyone praised ____ | The ***xyz*** did a miracle, and everyone praised ____ | The ***broadcaster*** did a miracle, and everyone praised ____ |

Table 1: Variation of Clean and Corrupted Samples (i.e. $C_1$, $C_2$) Used for Different Bias Setup. **DSS** stands for Demographic Sensitive Structure, **GSS** stands for Gender Sensitive Structure.

localization. Firstly, we evaluate the extent to which the highest-scoring edges (obtained from EAP method described in Section 3) contribute to the overall value of the bias metric ($L$) compared to a baseline model (i.e. where all edges are used). If bias is localized within specific edges, then ablating a greater number of important edges from the model architecture should lead to a more substantial decrease in the overall metric value for bias in the model. Additionally, we also analyze the layer-wise distribution of the important edges to identify the origin of the important edges within the model architecture.

### 3.4 Stability of Important Edges

Here, our primary goal is to explore the stability of important edges identified using the EAP approach described in Section 3. By stability, we refer to how much the important edges change with respect to different criteria. We primarily focused on three different criteria. Each of them is described as follows.

**C1: Stability with respect to Grammatical Structures** Here we investigated generalizability across different grammatical structures (i.e. DSS1, DSS2, GSS1 and GSS2 described in Table 1). Even if the grammatical structure changes, the gender/demographic bias present in a sentence is of the same nature. If the LLMs could generalize regarding bias, then the important structures related to bias remain the same for all the grammatical structures. To investigate this, we tested the different grammatical structures mentioned in Table 1. The difference between the grammatical structures for both demographic and gender bias is that, in demographic bias, the semantic meanings of the two variations are almost similar. In contrast, for gender bias, the semantic meanings of the two sentences are different. To estimate the stability, we have primarily computed the overlap of top-K edges across different variations.

**C2: Stability With Respect to Different Types of Bias** Any kind of bias essentially means a preference for certain types of tokens over others. Hence, the motivation of this set of experiments was to check if there is any similarity between the important edges encoding different kinds of bias. In our research scope, we looked into only demographic and gender bias.

**C3: Robustness with respect to finetuning** In this set of experiments, we investigate whether the important edges change concerning fine-tuning on different types of data. Here, we primarily did two different sets of experiments. Several studies Zmigrod et al. (2019); Han et al. (2024); Morerio et al. (2024) have demonstrated that targeted fine-tuning with augmented data—specifically curated to address the bias category of interest—can effectively mitigate biases in language models. In that direction, for the first set of experiments, we finetuned the model with a dataset where positive things were said about the countries/professions where there was negative demographic bias and vice versa. The objective of this experiment was to check whether debiasing the model by fine-tuning changes the important edges for generating sentences of the same grammatical structure. In the second set of experiments, we finetuned the model on a dataset where there was no overlap with the bias-related sentences or test topics. The bias of the model doesn't change after this. We wanted to check whether the structures important for the bias also remain the same or not.

### 3.5 Leveraging Important Edges for Debiasing LLMs

Existing debiasing techniques predominantly focus on data augmentation, modifications to the training objective, or fine-tuning the model. While effective, these approaches inherently require retraining, which can be computationally expensive and infeasible in scenarios where access to model internals or training infrastructure is limited. To overcome this limitation, we propose a novel inference-time debiasing strategy that leverages the important edges identified via the EAP method (Section 3). Our approach involves

substituting the activations (logits) along bias-associated edges with those obtained from a corrupted version of the input, while preserving clean activations along the remaining edges. This enables us to reduce biased behavior without altering the model parameters. Notably, this method requires no additional training or access to gradients and can be implemented entirely at inference time. To implement this technique, a corrupted variant of the original input is provided alongside the clean input. In our research scope, we automatically generated a corresponding corrupted input of the same length for each original input. The details of this setup are described in Appendix A.6.

Prior work Kaneko et al. (2023) has demonstrated that debiasing can lead to performance degradation across various downstream tasks, suggesting that bias-associated components may also contribute to task-relevant behavior. To assess the broader impact of our debiasing method, we evaluated its effectiveness on a range of standard NLP tasks. Specifically, we apply the corrupted logits approach to multiple settings to investigate whether the mitigation of bias incurs a trade-off in task performance.

## 4    Experiment Setup

For demographic and gender bias-related experiments, we used two different types of datasets. The ***Demographic Bias*** dataset used in our experiment is from an existing study in Narayanan Venkit et al. (2023). According to the findings in Narayanan Venkit et al. (2023), GPT-2 and Llama-2 models exhibit nationality bias in text generation. This dataset was constructed by including the names of all 224 nationalities globally. The study in Narayanan Venkit et al. (2023) identified a negative bias towards certain nationalities among these 224 nationalities. Our research specifically targets two distinct sentence structures (Described in Table 1) that mention nationality. We then assess the bias in GPT-2 or Llama-2 by examining the text completion of these sentence structures. The detailed descriptions about the model architecture of GPT-2 and Llama-2 models used in our experiment are given in Appendix A.2.

To compute demographic bias in a text completion setup, we get the top-k next token predictions for every sentence (k = 10 in our case, explained in detail in Appendix A.1) by the model and for every sentence we concatenate each of these predictions separately to the sentence and check the sentiment scores of resulting sentences using **Distilbert-base-uncased** model HF Canonical Model Maintainers (2022). If the sentiment of the sentence is positive, then we assume that the token predicted by the model was positive and vice versa. For quantitatively computing the bias of the dataset 5, we used a metric (similar to $L_1$ in Equation 2) which computes the difference between positive and negative probabilities. Consequently, the Demographic Bias metric should be positive in the case of a Positive-Bias Dataset and negative in the case of a Negative-Bias Dataset. Using the methods described above, we divide our demographic dataset into two categories: Positive-Bias dataset and Negative-Bias dataset. If the sum of the probabilities of next token predictions (for which the sentiment is positive) is greater than or equal to the sum of the probabilities of next token predictions (for which the sentiment is negative) for a sentence, it is classified into the Positive-Bias dataset and vice versa.

To understand ***Gender Bias*** in models, we used the set of 320 professions chosen and annotated from Bolukbasi et al. (2016b). It is an exhaustive list of gender-specific and gender-neutral terms extracted from the w2vNEWS word embedding dataset Bolukbasi et al. (2016a). The dataset was formed on similar grounds to the Demographic one, using sentence structure prompts of mainly two types as described in Table 1. Using the gender dataset, we compute the bias in a similar way to demographic bias. The only difference is rather than using the sentiment score of a sentence, we compare the predicted token with the exhaustive set of male and female-specific common words as mentioned in Bolukbasi et al. (2016b). Each predicted word is then assigned to three groups: **Male-Stereotypical(MS), Female-Stereotypical(FS), and Gender-Neutral(GN)**. Similar to demographic dataset, we divide our gender dataset into two categories: Male-Biased dataset and Female-Biased dataset. If the sum of MS probabilities (of next token predictions) is greater than or equal to the sum of FS probabilities (of next token predictions) for a sentence, it is classified into the Male-Biased dataset and vice versa. Detailed dataset statistics for both demographic and gender bias and the corresponding bias estimations are given in Table 5 in Appendix A.1.

| Bias Type | Configuration | Change in Metric | | | | Bias Type | Change in Metric | | | |
|---|---|---|---|---|---|---|---|---|---|---|
| | | GPT-2 Small | Llama-2 | Qwen-2-0.5b | Gemma | | GPT-2 Small | Llama-2 | Qwen-2-0.5b | Gemma |
| $DSS1_{pos}$ | $L_1, C_1$ | 0.2987 | 0.2218 | 0.2322 | 0.2123 | $GSS1_{pos}$ | 0.1982 | 0.1834 | 0.2164 | 0.2678 |
| $DSS1_{pos}$ | $L_1, C_2$ | 0.2691 | 0.2441 | 0.0462 | 0.6021 | $GSS1_{pos}$ | 0.1599 | **0.2261** | 0.1812 | 0.2153 |
| $DSS1_{pos}$ | $L_2, C_1$ | 0.2331 | 0.2981 | 0.2081 | 0.3267 | $GSS1_{pos}$ | 0.1498 | 0.2418 | 0.2315 | 0.2179 |
| $DSS1_{pos}$ | $L_2, C_2$ | **0.1982** | **0.2007** | **0.0148** | **0.1052** | $GSS1_{pos}$ | **0.0259** | 0.2426 | **0.1321** | **0.2001** |
| $DSS1_{neg}$ | $L_1, C_1$ | 0.2887 | 0.2583 | 0.2233 | 0.2682 | $GSS1_{neg}$ | 0.2276 | 0.2939 | 0.2461 | 0.2718 |
| $DSS1_{neg}$ | $L_1, C_2$ | 0.1553 | 0.2804 | 0.3012 | 0.2189 | $GSS1_{neg}$ | 0.2558 | 0.2919 | 0.2322 | 0.2123 |
| $DSS1_{neg}$ | $L_2, C_1$ | **0.0553** | 0.6993 | 0.2351 | 0.3416 | $GSS1_{neg}$ | 0.1930 | 0.1866 | 0.2531 | 0.2541 |
| $DSS1_{neg}$ | $L_2, C_2$ | 0.0758 | **0.1520** | **0.1015** | **0.2011** | $GSS1_{neg}$ | **0.2333** | **0.1511** | **0.1211** | **0.1301** |

Table 2: Variation in change in Metric value under different metric (i.e. $L_1$ and $L_2$) and corruption configurations $C_1$ and $C_2$. Each change value is normalized with respect to the original metric value produced by the model. For each category of bias (i.e. $DSS1_{pos}$, $DSS1_{neg}$, $GSS1_{pos}$, $GSS1_{neg}$) the lowest change in metric value is boldfaced.

**Implementation Details** We primarily experimented with the **Hooked-Transformer** from **Transformer-Lens** repository[2] which offers a modular and transparent framework for studying the internal mechanisms of transformer models like GPT-2 and Llama-2. For the finetuning experiments, the pretrained GPT-2 and Llama-2 were finetuned to examine the change in underlying circuit with respect to two different types of datasets. In one variation, the model was fine-tuned on a ***Positive Dataset*** where all countries were given positive tokens (for gender bias case it is male gender bias) so that the finetuned model is biased toward positive sentiment irrespective of the nationality. The goal was to examine whether this fine-tuning approach alters the key edges responsible for generating sentences with similar grammatical structures. In the second variation, the model was fine-tuned on a ***Shakespeare Dataset***, which consisted of Shakespeare text completely unrelated to the test sentences or topics. This experiment aimed to evaluate whether fine-tuning on entirely different content affects the edges contributing to bias. We conducted all the experiments in a computing machine having two A100 GPUs.

Based on the above discussion we would like to note that we will use the notations $DSS1$, $DSS2$ to describe demographic bias in the grammatical structures described in Table 1. Similarly $GSS1$ and $GSS2$ notations will be used to describe gender bias in the grammatical structures described in Table 1. We will use the terminology $DSS1_{pos/neg}$ ($GSS1_{pos/neg}$) to describe positive (male) or negative (female) bias.

## 5 Results

As described in Section 3, we show the effect of variations of corruption technique (described in Table 1) and metric $L$ (described in Section 3.1) in EAP for bias identification in Table 2. The 'Change in Metric' column in Table 2 computes the difference in metric value when all the edges are used vs. only the important edges are used. From Table 2, it can be observed that the difference in metric value (i.e. change in Metric column in Table 2) exhibits minimal change across variations in the corrupted structures (i.e. $C_1$ and $C_2$). However, when comparing $L_1$ and $L_2$, it is evident that the absolute difference in metric values are mostly much smaller (except for $DSS1_{pos}$ dataset in Llama-2) for $L_2$ than for $L_1$. Smaller change implies that set of important edges identified by EAP performs almost similarly as the whole model where all the edges have been cosidered. Consequently, we used $L_2$ and $C_2$ as metric for all the remaining demographic and gender bias analysis experiments. We also show the top 3 edges corresponding to different types of bias across different types of models in Table 7 in Appendix A.4.

Figure 2 and 3 show the results for localization of bias-related components in GPT-2 and Llama-2. In Figure 2, we report, for each layer in the LLM, the number of important edges—displaying only those layers that contain more than 20% of the total important edges to highlight the non-uniform distribution across the model architecture. We can observe from Figure 2 that for all the models (i.e. GPT-2 Small, GPT-2 Large and Llama-2) only a few layers contribute to the important edges for bias (i.e. For GPT-2 Small it is layer $2-6$, for GPT-2 Large it is primarily $9, 10, 20, 34, 35$ and for Llama-2 it is $0-11$ and $30-31$). The observation from the Figure 2 confirms that bias is encoded in certain layers only. Since there is a significant overlap between important edges for $DSS1_{pos}$ ($GSS1_{pos}$) and $DSS1_{neg}$ ($GSS1_{neg}$), in Figure 2 we only show the edge distribution for $DSS1_{pos}$ and $GSS1_{pos}$. It can be observed from Figure 3 that within 40% of the top edges for GPT-2 Small and Llama-2 models, the metric value drops by more than 90%. For GPT-2 Large, it

---

[2] https://github.com/TransformerLensOrg/TransformerLens

requires 60% of the edges to drop the metric value by 90%. Consequently, we can say that demographic and gender bias are encoded within a few edges in GPT-2 Small, GPT-2 Large, and Llama-2.

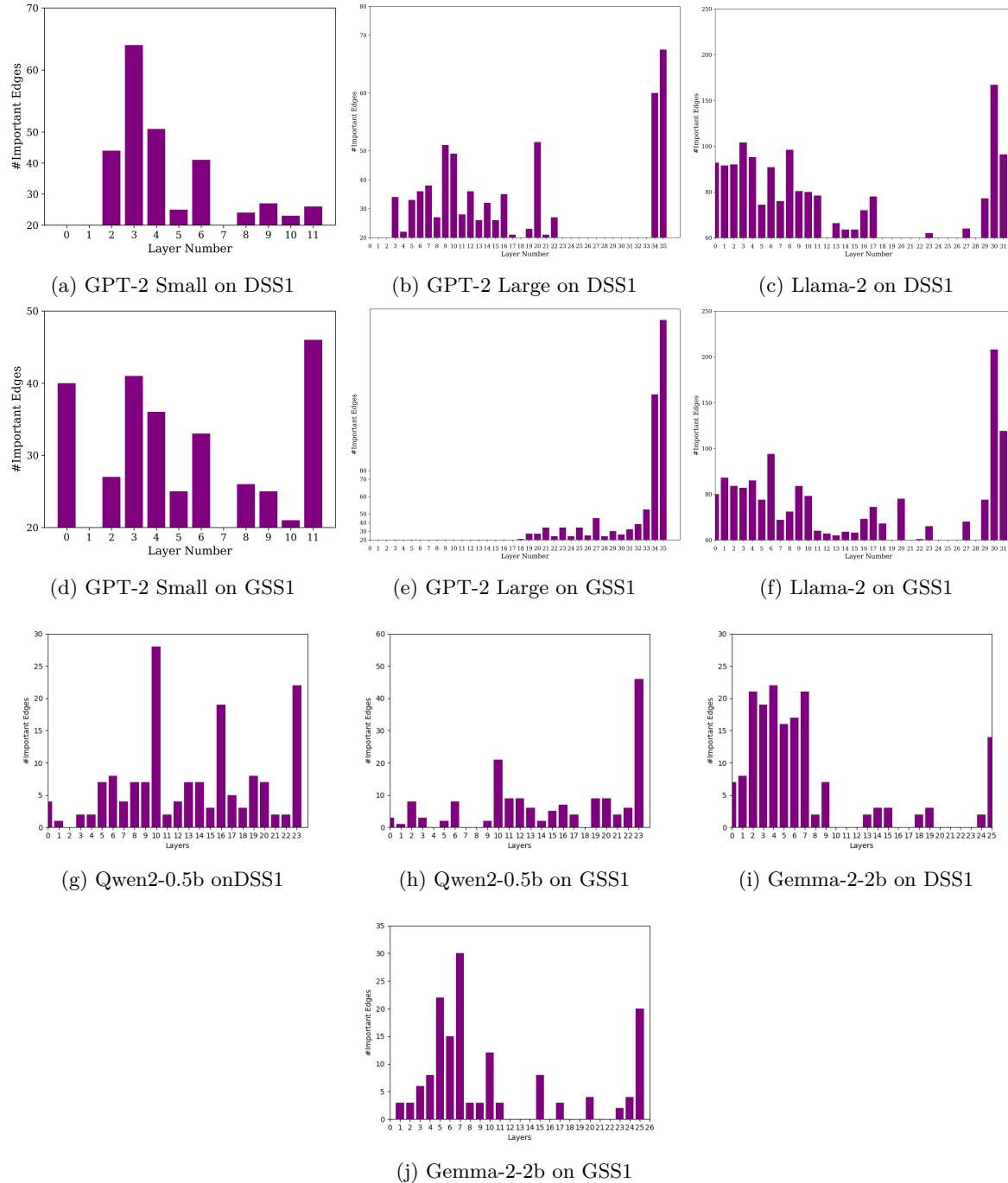

Figure 2: Layerwise important edge distribution for demographic bias (DSS1) and Gender bias (GSS1) across different models (i.e. GPT-2 Small, GPT-2 Large, LLAMA-2, Qwen 2-0.5 b and Gemma2-2b from left to right).

Figure 4 and 5 illustrates the generalizability and robustness of the edges identified using the EAP approach. In Figure 4, the three confusion matrices show the overlap among the top $k$ edges for both demographic and gender bias. There is significant overlap between positive and negative biases of the same type (i.e., demographic or gender). However, there is minimal to no overlap between demographic and gender biases. This pattern is consistent across GPT-2 (Small & Large), Llama-2, Qwen2-0.5b and Gemma2-2b models.

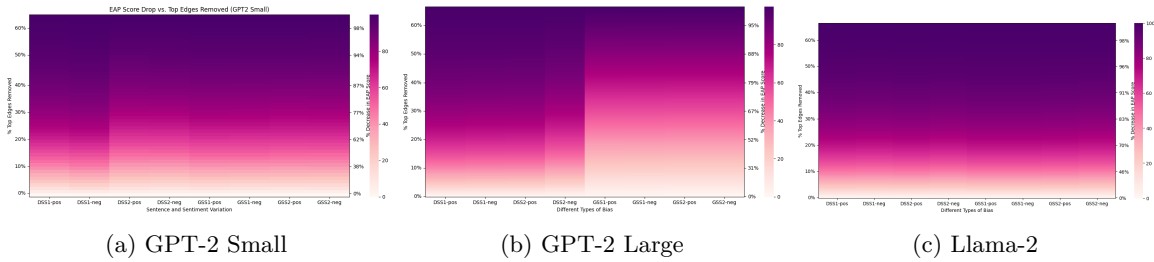

(a) GPT-2 Small        (b) GPT-2 Large        (c) Llama-2

Figure 3: Drop in $L_2$ value with % of Edge Ablation from GPT-2 Small, GPT-2 Large and Llama-2 across different configurations (i.e. $DSS1_{pos}$, $DSS2_{pos}$, $DSS1_{neg}$, $DSS2_{neg}$, $GSS1_{pos}$, $GSS2_{pos}$, $GSS1_{neg}$, $GSS2_{neg}$).

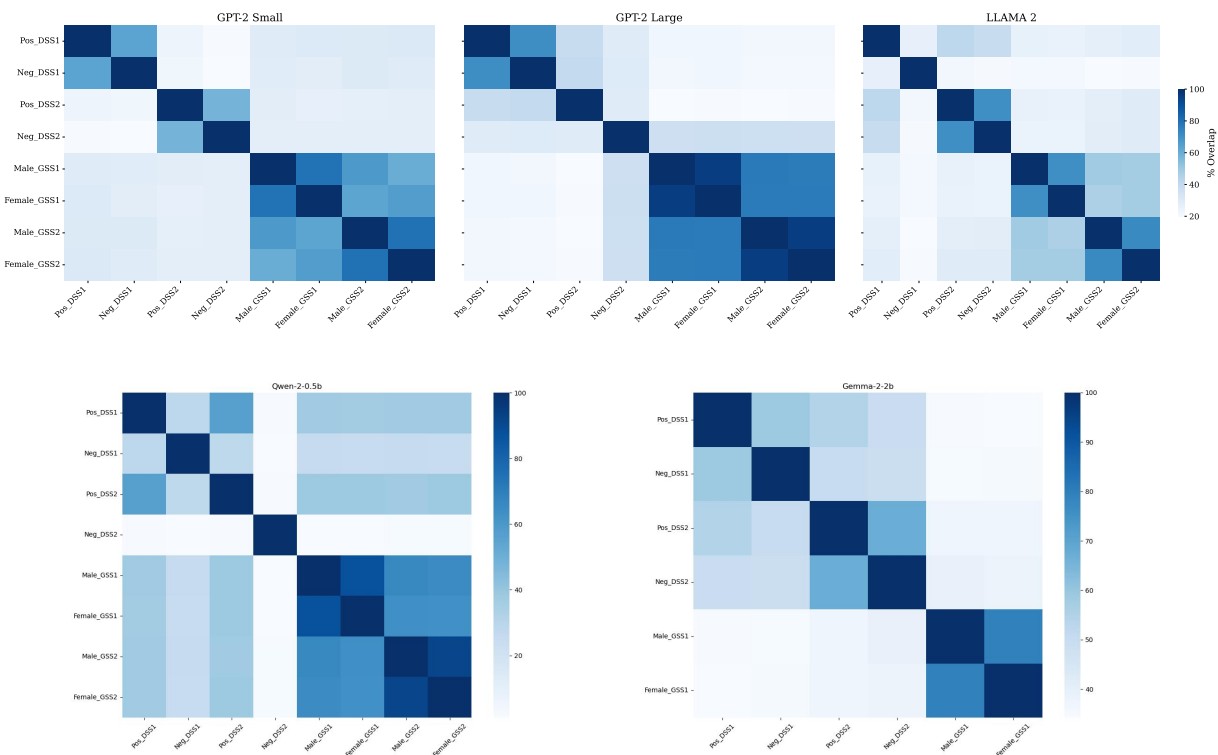

Figure 4: Plot showing overlap of top edges by EAP scores for GPT-2 Small, GPT-2 Large, Llama-2, Qwen2-0.5b and Gemma2-2b over different bias and sentence structure variation.

These results suggest that the similarity and dissimilarity patterns across different circuit variations are consistent across models. It is also shown that there is very less overlap between the edges responsible for demographic and gender bias. An interesting observation from Figure 4 is that, for demographic bias under grammatical variation (i.e., DSS1 and DSS2), the sets of important edges exhibit minimal overlap. In contrast, gender bias under similar variations (i.e., GSS1 and GSS2) reveals a substantial degree of overlap, suggesting greater structural consistency in how gender-related information is represented within the model. We have shown a similar analysis with another causal intervention technique Marks et al. (2025) in Appendix A.8.

To evaluate the stability of identified edges (i.e. described in Section 3.4), we analyzed the overlap between edges in the pre-trained model and the fine-tuned model in Figure 5. Fine-tuning was performed on two different datasets (i.e. *Positive* and *Shakespeare* as described in Section 4). Interestingly, as shown in the figure, fine-tuning with different datasets caused noticeable changes in the circuit components, and this observation held true for both small models (GPT-2 Small) and large models (GPT-2 Large and Llama-2).

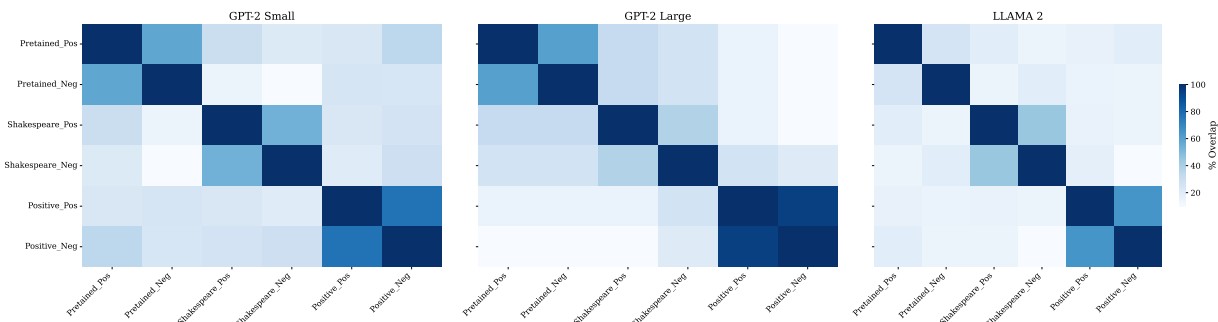

Figure 5: Plot Showing Overlap of top edges by EAP scores for Untuned vs Finetuned GPT-2 Small, GPT-2 Large, and Llama-2 in $DSS1$ (Demographic) configuration. Pretrained_Pos, Pretrained_Neg show positive or negative bias in pretrained LLMs. Shakespeare_Pos and Shakespeare_Neg denote positive or negative biased models finetuned on the Shakespeare dataset. Positive_pos and Positive_neg show the underlying circuit in the pretrained model finetuned on a positive dataset.

From Table 3, we can see that corrupting the top edges responsible for bias reduced bias in the original model in most of the cases, except for GPT-2 Large in DSS1. We used the top 400, 1000 and 3000 edges from GPT-2 Small, GPT-2 Large, and Llama-2, respectively. We also wanted to investigate whether this edge corruption also affects the performance of the model in other NLP tasks. Consequently, Table 3 shows the performance on two NLP tasks: CoNLL-2003 Sang & De Meulder (2003), a named entity recognition benchmark, and CoLA Warstadt (2019), a linguistic acceptability judgment task for all the models. Table 3 shows a decrease in performance in CoLA and CoNLL-2003 tasks across all the models. This reduction shows that bias-related edges have an overlap with different language understanding tasks. This differential impact across tasks hints at a hierarchical organization of linguistic knowledge within the network, where certain capabilities are more deeply integrated into these high-influential edges than others. Table 4 shows the performance of debiased GPT2 Drechsel & Herbold (2025) and Lllama3.2 Drechsel & Herbold (2025) in CoNLL-2003 Sang & De Meulder (2003), CoLA Warstadt (2019) and NER-CONL2003. It can observed that the decrase in performance in the above mentioned tasks is much more compared to our proposed debiasing approach in Table 3.

| | | GPT-2 Small | | | GPT-2 Large | | | Llama-2 | | | Qwen-20.5b | | | Gemma | | |
|---|---|---|---|---|---|---|---|---|---|---|---|---|---|---|---|---|
| | Model | $\delta$ Bias | CoLA | NER | $\delta$ Bias | CoLA | NER | $\delta$ Bias | CoLA | NER | $\delta$ Bias | CoLA | NER | $\delta$ Bias | CoLA | NER |
| DSS1 | Proposed | 35.88%↓ | 22.6%↓ | 20.40%↓ | 8.89%↑ | 3.09%↓ | 6.03%↓ | 9.16%↓ | 2.10%↓ | 0.01%↓ | 41.02%↓ | 3.25%↓ | 0.06%↓ | 36.01%↓ | 2.15%↓ | 1.01%↓ |
| DSS2 | Proposed | 30.37%↓ | 18.13%↓ | 12.63%↓ | 71.30%↓ | 4.67%↓ | 3.37%↓ | 35.40%↓ | 0.01%↓ | 0.01%↑ | 35.56%↓ | 1.25%↓ | 0.01%↓ | 25.61%↓ | 1.91%↓ | 0.02%↓ |
| GSS1 | Proposed | 21.85%↓ | 0.66%↓ | 2.70%↓ | 2.87%↓ | 6.43%↓ | 0.18%↓ | 28.84%↓ | 5.70%↓ | 6.28%↓ | 25.31%↓ | 1.25%↓ | 1.21%↓ | 31.01%↓ | 1.15%↓ | 0.08%↓ |
| GSS2 | Proposed | 19.86%↓ | 2.55%↓ | 2.83%↓ | 1.15%↓ | 10.00%↓ | 1.23%↓ | 25.00%↓ | 7.08%↓ | 3.90%↓ | 23.12%↓ | 2.03%↓ | 1.16%↓ | 22.16%↓ | 1.03%↓ | 0.07%↓ |

Table 3: $\delta$ Bias shows the change in Bias between the output produced by a pretrained model and the model for which bias responsible edges were corrupted. ↓ shows decrease in bias and ↑ shows increase in bias. We also show performance change for CoLA and NRE-CoNL2023.

| | | Debiased GPT-2 | | | Debiased Llama | | | Debiased Qwen | | | Debiased Gemma | | |
|---|---|---|---|---|---|---|---|---|---|---|---|---|---|---|
| Bias Type | Method | $\delta$ Bias | CoLA | NER-CoNLL2003 | $\delta$ Bias | CoLA | NER-CoNLL2003 | $\delta$ Bias | CoLA | NER-CoNLL2003 | $\delta$ Bias | CoLA | NER-CoNLL2003 |
| GSS1 | Pretrained | 37.21%↓ | 28.9%↓ | 25.30%↓ | 9.22%↓ | 5.61%↓ | 4.29%↓ | - | - | - | - | - | - |
| GSS1 | Chintam et al. (2023) | 20.41%↓ | 26.15%↓ | 14.24%↓ | 21.11%↓ | 9.12%↓ | 6.24%↓ | 23.95% | 3.51% | 4.23% | 29.89% | 4.15% | 1.53% |
| GSS1 | Xu et al. (2025) | 22.13%↓ | 31.22%↓ | 28.21%↓ | 30.22%↓ | 18.32%↓ | 7.55%↓ | 25.31% | 4.13% | 6.21% | 31.59% | 2.33% | 2.14% |

Table 4: Performance of Gender debiased GPT2, Llama, Qwen2-0.5b and Gemma2-2b model. Only the pretrained experiment in Llama is done using Llama3.2 model. However rest of the Llama experiments are done on Llama2 only. The first row shows results for models trained as a debiased model.

## 6 Conclusion

In this work, we investigated how bias is structurally embedded within the architectures of GPT-2 Small, GPT-2 Large, LLaMA-2, Qwen2-0.5b, Gemma 2-2b using tools from mechanistic interpretability. Our analysis revealed that, across model scales and architectures, circuits responsible for different categories of bias—such

as demographic and gender bias—are largely disjoint, indicating that the underlying representations of bias are modular and specialized. Through targeted interventions on specific edges identified via our causal metrics, we demonstrated that it is possible to attenuate bias in model outputs without retraining. However, we also observed that these structural manipulations can negatively impact the model's performance on unrelated NLP tasks, such as named entity recognition and natural language inference. Our findings underscore a fundamental trade-off between debiasing and general task performance, and point to the need for more selective interventions that can isolate bias-related functionality while preserving broader model competence.

**Limitations** One of the limitations of this project is that we focus specifically on demographic and gender bias in this work. It can be observed from our findings that for different types of bias, the underlying circuit responsible for it will be generally different. Consequently, for other biases, the circuits obtained from this work may not be applied.

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

# A   Appendix

## A.1   Description for Biased Dataset

We did an experiment varyting the number of tokens used to computed bias for GPT-2 and Llama-2 and found that beyond 10 tokens the amount of bias present in the model doesn't change that much. Hence we opted for only using 10 tokens to compute bias in LLMs. Table 5 shows the bias computed for each model on different datasets. It also shows the number of instances present in each variation of the dataset along with average length of the input sentences. The temperature is set to 1 for the generation from all models.

## A.2   LLMs Explored

We used three different models for our experiments. Each one of them is described as follows.

**GPT-2 Small** This is a 85M parameter model with 12 layers and 12 attention heads per layer. The model has a dimension of 768 and vocab size of 50257. It uses GELU as its activation function Radford et al. (2019). In its computational graph we have 158 nodes and 32491 edges.

**GPT-2 Large** This is a larger version of GPT having 708M parameters, with 36 layers and 20 attention heads per layer. This one has a dimension of 1280 and vocab size of 50257. Similar to the smaller version it uses GELU as its activation function Radford et al. (2019). In its computational graph we have 758 nodes and 810703 edges.

**Llama-2** This is a 6.5B parameter model with 32 layers and 32 attention heads per layer. The model has a dimension of 4096 and vocab size of 32000. Unlike GPT-2 versions it employs SiLU as its activation function Touvron et al. (2023). In its computational graph we have 1058 nodes and 1592881 edges.

| Model | Struct Type | Bias Category | #Samples | Length | Bias Metric |
|---|---|---|---|---|---|
| GPT-2 Small | DSS1 | Positive | 92 | 7.65 | 0.659 |
| | DSS1 | Negative | 132 | 6.72 | 0.753 |
| | DSS2 | Positive | 141 | 9.48 | 0.637 |
| | DSS2 | Negative | 83 | 9.06 | 0.633 |
| | GSS1 | Male | 293 | 11.69 | 0.816 |
| | GSS1 | Female | 27 | 11.74 | 0.751 |
| | GSS2 | Male | 291 | 10.69 | 0.837 |
| | GSS2 | Female | 29 | 10.75 | 0.787 |
| GPT-2 Large | DSS1 | Positive | 139 | 7.65 | 0.785 |
| | DSS1 | Negative | 85 | 6.85 | 0.742 |
| | DSS2 | Positive | 69 | 9.65 | 0.652 |
| | DSS2 | Negative | 155 | 9.18 | 0.762 |
| | GSS1 | Male | 290 | 11.68 | 0.889 |
| | GSS1 | Female | 30 | 11.8 | 0.823 |
| | GSS2 | Male | 274 | 10.66 | 0.847 |
| | GSS2 | Female | 46 | 10.91 | 0.786 |
| Llama-2 | DSS1 | Positive | 216 | 7.0 | 0.853 |
| | DSS1 | Negative | 8 | 6.125 | 0.718 |
| | DSS2 | Positive | 216 | 10.0 | 0.813 |
| | DSS2 | Negative | 8 | 9.0 | 0.665 |
| | GSS1 | Male | 298 | 13.33 | 0.861 |
| | GSS1 | Female | 22 | 13.36 | 0.797 |
| | GSS2 | Male | 291 | 13.31 | 0.864 |
| | GSS2 | Female | 29 | 13.55 | 0.760 |
| Qwen-2.05b | DSS1 | Positive | - | 7.0 | 0.718 |
| | DSS1 | Negative | - | 6.125 | 0.652 |
| | DSS2 | Positive | - | 10.0 | 0.798 |
| | DSS2 | Negative | - | 9.0 | 0.665 |
| | GSS1 | Male | - | 13.33 | 0.759 |
| | GSS1 | Female | - | 13.36 | 0.812 |
| | GSS2 | Male | - | 13.31 | 0.768 |
| | GSS2 | Female | - | 13.55 | 0.660 |
| Gemma | DSS1 | Positive | - | 7.0 | 0.845 |
| | DSS1 | Negative | - | 6.125 | 0.792 |
| | DSS2 | Positive | - | 10.0 | 0.851 |
| | DSS2 | Negative | - | 9.0 | 0.693 |
| | GSS1 | Male | - | 13.33 | 0.782 |
| | GSS1 | Female | - | 13.36 | 0.695 |
| | GSS2 | Male | - | 13.31 | 0.861 |
| | GSS2 | Female | - | 13.55 | 0.731 |

Table 5: Dataset Statistics for Different types of Bias Across Different Models. *#Samples*- Number of Samples, *Length*- Avg Length of the sentence. *Bias metric* - Normalized probability of Positive words per example of Postive Bias Category (and similarly the Normalized probability of Negative words per example for Negative Bias Category). Similar strategy for Gender Bias.

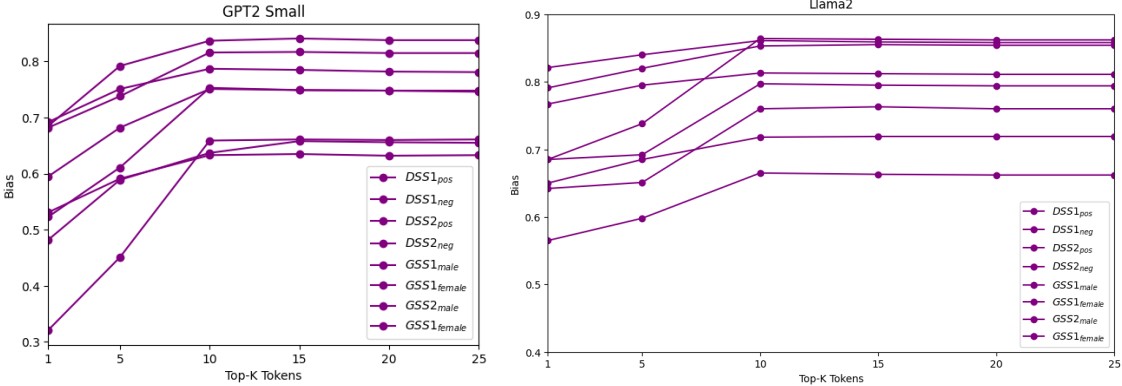

Figure 6: Bias Score Variation With Respect to Top-K values across GPT2 and Llama2

## A.3    Baseline Scoring

We have the option to calculate the baseline scores for both positive-bias and negative-bias datasets via two methods i.e., evaluate-baseline scoring and evaluate-graph scoring. The evaluate-baseline function calculates the difference in probabilities between positive and negative next-token predictions and averages it over the

| Node | Description | GPT-2 Small | GPT-2 Large | Llama-2 |
|---|---|---|---|---|
| input | Input Node | NA | NA | NA |
| logits | Logit Node | NA | NA | NA |
| m{i} | i$th$ MLP layer | $i \in (0-11)$ | $i \in (0-35)$ | $i \in (0-31)$ |
| a{i}.h{j} | j$th$ attention head in i$th$ attention layer | $i \in (0-11)$, $j \in (0-11)$ | $i \in (0-35)$, $j \in (0-19)$ | $i \in (0-31)$, $j \in (0-31)$ |
| a{i}.h{j}<x> x=[**q**]uery, [**k**]ey, [**v**]alue | j$th$ [x] attention head in i$th$ attention layer | $i \in (0-11)$, $j \in (0-11)$ | $i \in (0-35)$, $j \in (0-19)$ | $i \in (0-31)$, $j \in (0-31)$ |

Table 6: Nomenclature for Nodes in the computational graph of GPT-2 and Llama-2 (Edges described in Manual Edge Ablation Section in Appendix)

whole dataset. While as in evaluate-graph function we have the option of passing the argument of Graph (A Graph represents a model's computational graph. Once instantiated, it contains various Nodes, representing mostly attention heads and MLPs, and Edges representing connections between nodes. Each Node and Edge is either in the graph (circuit) or not; by default, all Node and Edge objects are included within the graph.), and in the process of calculating the baseline score using evaluate-graph function we need to pass the unaltered graph where no edge or node is ablated yet. Since we are going to use the evaluate-graph function when we ablate some edges, hence it is best to use evaluate-graph function for getting the baseline score which we are going to use to check the importance of edges.

## A.4 Finding Bias Circuits

We sort all the edges in the graph according to their scores and print the edges in the descending order with their respective scores for each positive-bias and negative-bias dataset. These scores are the respective edge scores and reveal the importance of the edges in the graph for propagating the respective bias. The more the score, the more important is the edge. The goal here was to ablate the Top N edges in the graph (N ranging from 1 to 10 in our case) and observe the variation in the evaluate-graph scores and compare it to the graph-baseline score. EAP Syed et al. (2024) eventually outputs a sorted list of edges where each edge is represented by the corresponding connecting nodes. In Table 6 we the description of different types of nodes referred in EAP. Thable 7 shows the top 3 edges obtained for different types of bias across different models (i.e. GPT2- Small, GPT2-large and Llama2).

## A.5 Selecting top-K edges

It was computationally infeasible to consider all the edges of a model to do all the analysis. Our primary object was to find the set of edges which were mostly responsible for bias in the model. Hence, we selected the top-k edges, where k represents the least number of edges retained in the model such that the resulting bias remains within 20% of the original model's bias value. In Figure 7, we varied the total number of edges from 600 to 3000 across different models. The red line in Figure 7 shows the 20% of the original bias value of the model.

## A.6 Creating Corrupted Edges for Downstream Tasks

For CoLA dataset we filter the sentence containing atleast one noun, duplicate the sentence and swap every noun token with "XYZ", preserving positions. For CoLA dataset we swap any two randomly chosen words in the sentence.

## A.7 Finetuning Experiment Setup

We conducted our experiments using the Hooked-Transformer framework from the TransformerLens repository Our base architectures were GPT-2 and Llama-2. Both models were evaluated in their pretrained form and subsequently fine-tuned on two distinct datasets.

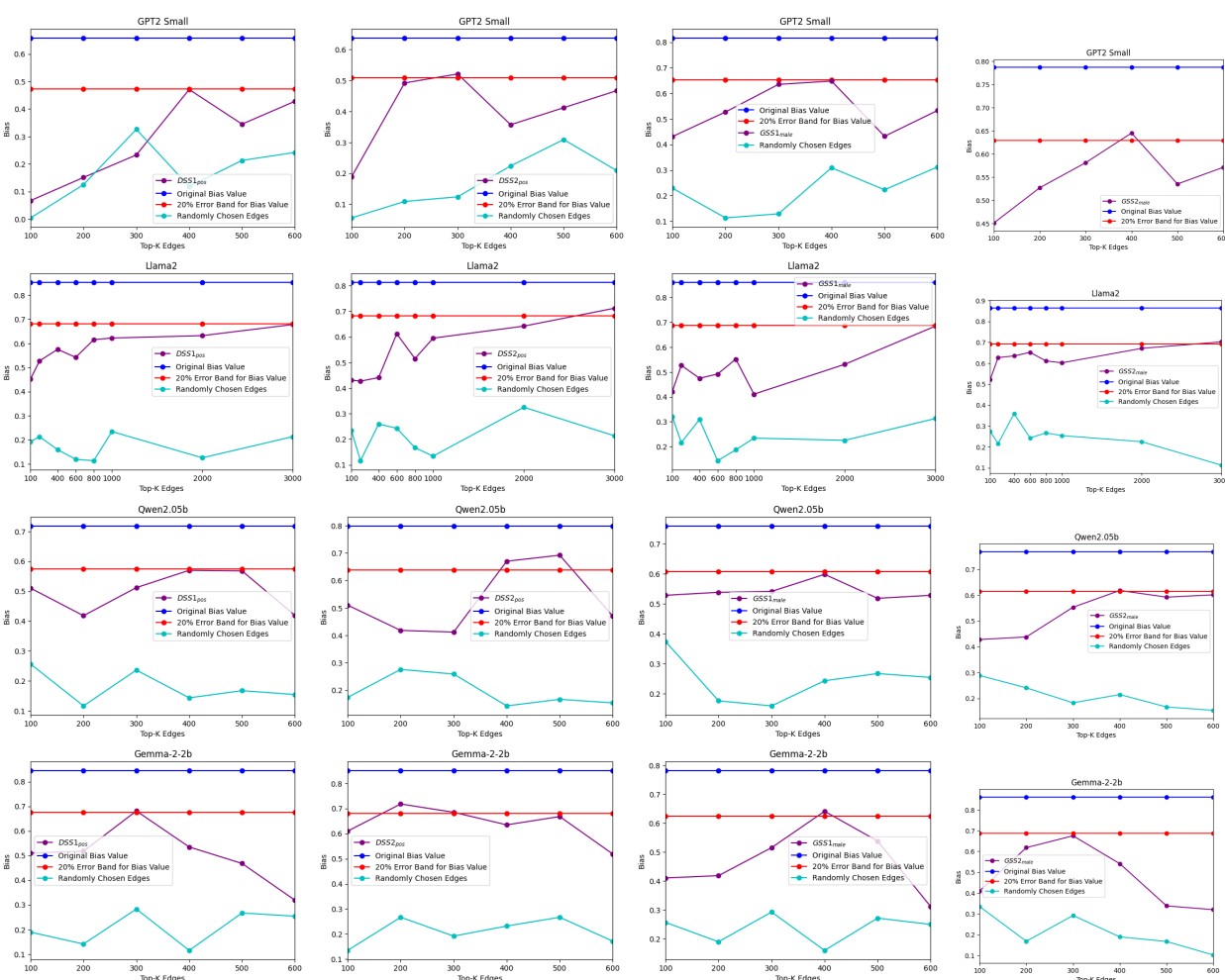

Figure 7: Variation of Top-K edges

| | | Top 1 | | Top 2 | | Top 3 | |
|---|---|---|---|---|---|---|---|
| **Model** | **Bias Type** | **Edge** | **Score** | **Edge** | **Score** | **Edge** | **Score** |
| GPT-2 Small | Demographic Positive(DSS1) | m11->logits | 0.3307 | m0->m2 | 0.1900 | m0->m1 | 0.1858 |
| GPT-2 Small | Demographic Negative(DSS1) | m11->logits | 0.2655 | m0->m1 | 0.1684 | m0->m2 | 0.1557 |
| GPT-2 Small | Demographic Positive(DSS2) | m11->logits | 0.1580 | m0->m2 | 0.1205 | m9->logits | 0.0921 |
| GPT-2 Small | Demographic Negative(DSS2) | m11->logits | 0.0720 | m0->m2 | 0.0658 | m0->m1 | 0.0527 |
| GPT-2 Large | Demographic Positive(DSS1) | m35->logits | 0.3183 | m33->m35 | 0.1732 | m33->logits | 0.1526 |
| GPT-2 Large | Demographic Negative(DSS1) | m35->logits | 0.3219 | m33->m35 | 0.1245 | a32.h2->logits | 0.1123 |
| GPT-2 Large | Demographic Positive(DSS2) | m35->logits | 0.2458 | m32->logits | 0.1226 | m33->m35 | 0.1070 |
| GPT-2 Large | Demographic Negative(DSS2) | m35->logits | 0.1151 | m32->logits | 0.0598 | m0->m4 | 0.0457 |
| GPT-2 Small | Gender Male(GSS1) | input->m0 | 0.0663 | input->a0.h5⟨k⟩ | 0.0561 | input->a0.h5⟨q⟩ | 0.0483 |
| GPT-2 Small | Gender Female(GSS1) | input->m0 | 0.0675 | input->a0.h5⟨k⟩ | 0.0539 | m11->logits | 0.0471 |
| GPT-2 Small | Gender Male(GSS2) | input->m0 | 0.0615 | input->a0.h5⟨k⟩ | 0.0602 | input->a0.h5⟨q⟩ | 0.0502 |
| GPT-2 Small | Gender Female(GSS2) | input->m0 | 0.0649 | input->a0.h5⟨k⟩ | 0.0575 | m11->logits | 0.0541 |
| GPT-2 Large | Gender Male(GSS1) | m35->logits | 0.0281 | a33.h11->logits | 0.0274 | m33->logits | 0.0261 |
| GPT-2 Large | Gender Female(GSS1) | m35->logits | 0.0206 | a33.h11->logits | 0.0203 | m33->logits | 0.0192 |
| GPT-2 Large | Gender Male(GSS2) | m35->logits | 0.0306 | a33.h11->logits | 0.0265 | a32.h2->logits | 0.0249 |
| GPT-2 Large | Gender Female(GSS2) | m35->logits | 0.0241 | a32.h2->logits | 0.0206 | a33.h11->logits | 0.0204 |
| Llama-2 | Gender Male(GSS1) | m31->logits | 0.2467 | m30->logits | 0.0560 | m30->m31 | 0.0497 |
| Llama-2 | Gender Female(GSS1) | m31->logits | 0.3141 | m30->logits | 0.1282 | m30->m31 | 0.0852 |
| Llama-2 | Gender Male(GSS2) | m31->logits | 0.2618 | m30->logits | 0.0907 | m30->m31 | 0.0542 |
| Llama-2 | Gender Female(GSS2) | m31->logits | 0.3308 | m30->logits | 0.1461 | m30->m31 | 0.1095 |

Table 7: Top 3 edges for Different Models and Different Bias

| | EAP Vs SAE Approach | | | | | | | |
|---|---|---|---|---|---|---|---|---|
| | Pos_DSS1 | Pos_DSS2 | Neg_DSS1 | Neg_DSS2 | Male_GSS1 | Male_GSS2 | Female_GSS1 | Female_GSS2 |
| GPT2 Small | 75% | 80% | 81% | 78% | 77% | 73% | 82% | 71% |
| GPT2 Large | 73% | 72% | 71% | 76% | 85% | 75% | 72% | 81% |
| Llama2 | 84% | 72% | 71% | 79% | 87% | 73% | 72% | 83% |

Table 8: Overlap percentage Between Important Edges Obtained from EAP and SAE.

The first fine-tuning variant used *Positive Dataset*, where all countries were paired with positive sentiment tokens. For the gender bias case, male-associated tokens were emphasized. This setup intentionally biased the model toward positive sentiment with the goal of testing whether such polarity shifts could serve as a debiasing mechanism while preserving the grammatical scaffolding of the model.

The second variant used *Shakespeare Dataset*, consisting of Early Modern English text. This fine-tune was designed not only to impose stylistic features but also to explore whether forcing the model into a highly structured linguistic domain could mitigate bias. By encouraging the models to adopt Shakespearean vocabulary and syntax, we aimed to examine whether stylistic adaptation reconfigures structural circuits in ways that contribute to debiasing or instead introduces new pathways specific to style.

**Training Details** Fine-tuning was performed with the AdamW optimizer using a learning rate of $10^{-4}$ and a batch size of 129. Each model was fine-tuned for 20 epochs on the respective datasets, with early stopping based on validation loss. Evaluation was conducted on held-out validation splits, and circuit changes were analyzed using attention weight inspection and edge attribution methods within TransformerLens.

## A.8 SAE Approach Results

SAE Marks et al. (2025) is another recent approach to find important components from a model similar to EAP Syed et al. (2024). In Figure 8, we have done similar experiments like Figure 2 using SAE. Figure 8 shows that there is not much overlap between demographic and gender bias in general. Table 8 shows the overlap between important edges obtained from EAP approach and SAE approach for all the different variations. It can be seen that the overlap is always more than 70%.

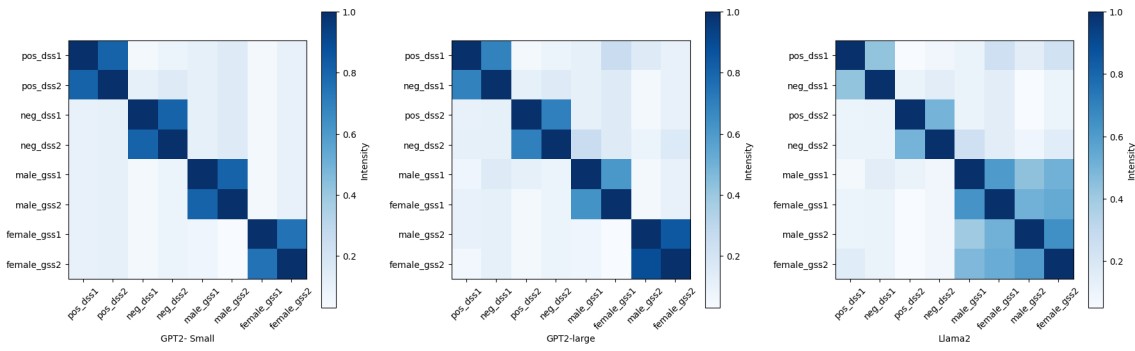

Figure 8: Overlap Results from SAE Approach

## A.9    Circuit Diagram

Figure 9 shows the circuit diagram for different types of bias (i.e. demographic and gender) obtained from EAP approach Syed et al. (2024). A circuit is a subgraph of a neural network[3].

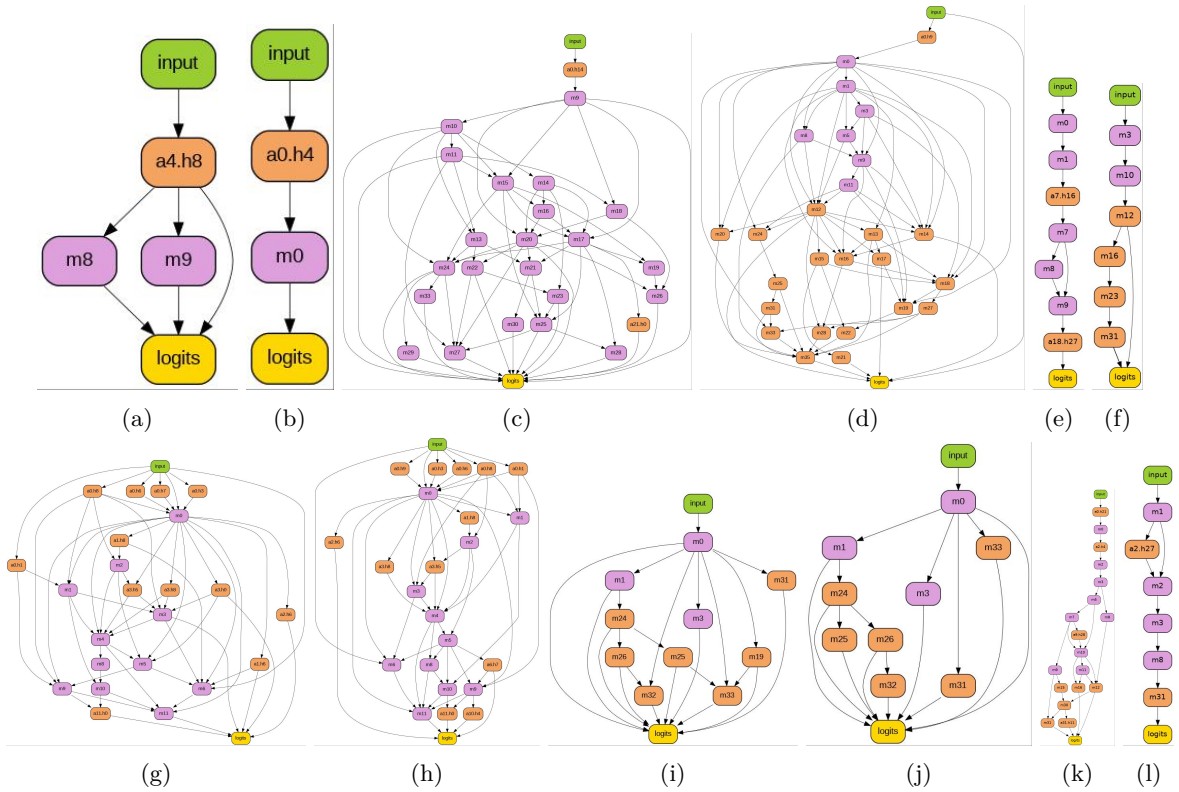

Figure 9: Circuit Diagram: (a) GPT-2-Small-DSS1-Positive, (b) GPT-2-Small-DSS2-Positive, (c) GPT-2-Large-DSS1-Positive, (d) GPT-2-Large-DSS2-Positive, (e) Llama-2-DSS1-Positive, (f) Llama-2-DSS2-Positive, (g) GPT-2-Small-GSS1-Positive, (h) GPT-2-Small-GSS2-Positive, (i) GPT-2-Large-GSS1-Positive, (j) GPT-2-Large-GSS2-Positive, (k) Llama-2-GSS1-Positive, (l) Llama-2-GSS2-Positive.

---

[3]https://distill.pub/2020/circuits/zoom-in/#glossary-circuit

