# OpenReview forum: "Dissecting Bias in LLMs: A Mechanistic Interpretability Perspective"
_TMLR — Accepted by TMLR_

### Review · Reviewer_1xiq · 2025-09-11

**Summary Of Contributions:**

The authors adopt a mechanistic interpretability(MI) perspective and examine demographic and gender bias in GPT-2 Small, GPT-2 Large, and Llama-2. This paper is motivated by the limited generalizability of prior debiasing approaches that primarily rely on fine-tuning or data augmentation; The authors share analysis on these previous methods as well. Using Edge Attribution Patching (EAP), they investigate whether biases are localized within distinct substructures of these models.


**Key findings:**
1. Demographic and gender bias are largely localized to certain edges and layers across models.
2. The location of bias-related edges in a network tends to shift  under perturbations (e.g., input variation, fine-tuning), rather than remaining consistent across conditions.
3.Debiasing by corrupting these edges reduces bias but also leads to a trade-off, lowering performance on downstream tasks. <br>



**Strengths:**
1. It addresses an important problem of debiasing LLMs, with a particular focus on whether bias can be localized within specific components of the model.
2. The paper leverages Edge Attribution Patching (EAP), an existing method for estimating the causal importance of edges with minimal computation (two forward and one backward pass).
3. The experiments are extensive. Experiments examine bias under diverse corrupted sample settings, evaluate stability across different fine-tuning settings, and compare results across models of different sizes and architectures.



**Weaknesses:**
1. Despite criticizing prior work for narrow scope, this paper also focuses only on two bias types and two architectures, leaving its generalizability uncertain.
2. The novelty of this paper appears limited. First, the methodology used—Edge Attribution Patching (EAP) and the L1/L2 bias metrics—relies on previously introduced techniques. Second, the central conclusion that bias is localized to specific components of LLMs is not entirely new, as similar claims have been made in prior work [1,2].
3. Some results are difficult to interpret, e.g., Table 2 shows only marginal differences without defining “minimal change,” and Figure 3’s localization evidence is less convincing for Llama-2 than for GPT-2 Small.
The study is limited to GPT-2 and Llama-2, raising concerns about the applicability of findings to more recent models.


[1] Jesse Vig et al. Investigating Gender Bias in Language Models Using Causal Mediation Analysis. In Advances in Neural Information Processing Systems (NeurIPS), 2020.

[2] Yi Yang et al. Bias A-head? Analyzing Bias in Transformer-Based Language Model Attention Heads. arXiv preprint arXiv:2311.10395, 2024.

**Audience:**

Yes

**Audience Explanation:**

The paper addresses a timely and important issue: understanding how bias is represented within LLMs through mechanistic interpretability. Even if limited in scope, the idea of bias localization at the edge/layer level and the use of EAP for analysis would be of interest to researchers working on LLM bias, interpretability, and safety.

**Broader Impact Concerns:**

No major broader impact concerns to note.

**Claims And Evidence:**

No

**Claims Explanation:**

The experiments are thoughtfully designed, but the evidence is not always convincing. The use of outdated models (GPT-2 and Llama-2) makes it unclear whether the findings would generalize to more recent LLMs. Some results (e.g., Table 2 and Figure 3) are ambiguous and not sufficiently justified, making the localization claim less robust. Additionally, the scope is limited to only two bias types and two architectures, which narrows the findings of this work despite the paper’s stated motivation of generalizability.  The description of the approach taken is also not specified enough to clearly understand pros and cons of the proposed mechanistic approach.

**Requested Changes:**

## 1. Details required for understanding overall experiments


**1-1. Corrupted samples**: Section 3.2 should provide more detail on how corrupted samples were constructed. While the paper cites prior work as the source of neutral terms, it remains unclear how many neutral replacements were sampled and what selection criteria were used. Clarifying this would improve transparency.


**1-2. Bias metrics**: Section 3.1 states that higher values of L1 indicate stronger bias. However, since L1 can be either positive or negative depending on the favored group, this phrasing is misleading. It would be more accurate to state that higher absolute values of L1 correspond to stronger bias, with the sign reflecting the direction of bias.


**1-3. Clarification on “Edge” Definition**: The definition of an “edge” is somewhat abstract. Please provide more concrete examples of what an edge represents in the computational graph (e.g., outputs of MLP blocks or attention heads).


**1-4. Definition of important edges**: In Section 5, Table 2 compares “important edges” with all edges. While the paper defines importance within the EAP framework, it does not specify the threshold criterion for classifying important edges. A more precise description is required to ensure reproducibility.


## 2. Missing technical details in methodology section


**2-1. EAP explanation**: EAP is a central methodology in this work and is claimed to be more efficient than naive ablation, requiring only two forward and one backward pass to compute importance scores for all edges. However, based on the formula provided, it is difficult to understand why such efficiency is achievable. While a full derivation may not be necessary here, a brief high-level explanation or more explicit reference to prior work would improve clarity for readers unfamiliar with EAP.


**2-2. Details on Gradient Computation for EAP with External Metrics**: The paper does not clearly explain how gradients of a loss defined by an external sentiment model (e.g., DistilBERT) are backpropagated to the LLM’s internal edges. Please clarify how differentiability is maintained across discrete token generation and whether any special techniques are used.


## 3. Miscellaneous


**3-1. Comparison with other debiasing methods**: The experiments only include one debiasing baseline. It would strengthen the paper to either compare with other established debiasing techniques or clarify why such comparisons were omitted.


**3-2. References**: On page 3, Syed et al. (2023) and Syed et al. (2024) are cited separately, but they appear to be essentially the same work on attribution patching (same authors and overlapping content). Unless there is a substantive difference in how these two papers are intended to be used as references, they should be combined into a single citation.

---

> ### Author Response · Authors · 2025-10-12
>
> We would like to thank the reviewer for the detailed comments. We have considered all your suggestions and submitted a revised draft. Our response to your **requested changes** is as follows.
>
> 1-1 **Corrupted Samples Creation:**  We would like to clarify that the corrupted samples were manually constructed. Each clean instance has one corresponding  corrupted version maintaining identical token length similar to STR strategy cited. For the demographic dataset, we replaced country names with a neutral placeholder token (e.g., “Abc” or a neutral country). Similarly, for the gender-bias dataset, we replaced profession-specific terms with a neutral placeholder (e.g. Xyz or a neutral profession term). The choice of neutral terms was guided by examining the model’s output distributions: we selected terms for which the predicted probabilities of next positive and negative tokens were approximately equal, ensuring that they did not introduce unintended bias. Table 1 shows samples of the corrupted sentences. For each clean instance there was only one corrupted sample.
>
> 1-2 **Bias Metric Value:** Thanks for pointing this out about L1 value. Yes absolute value is what matters for L1. We have updated the wordings in Section 3.1.
>
> 1-3 **Clarification on “Edge” Definition:** Table 6 in the Appendix describes the different types of nodes (i.e. Input nodes, logit nodes, MLP layer output, Attention head). An “edge” represents information flow and connection between a pair of nodes (e.g., MLP → logit node). Thus, an edge corresponds to a directed computational connection in the forward pass of the transformer. Figure 1 provides illustrative examples of such edges.
>
> 1-4 **Definition of Important Edges:** Figure 7 illustrates the effect of retaining only the top-k edges on the model’s bias. From this figure, we determined the value of k for which the model’s bias remained within 20\% of the bias observed in the original model (indicated by the blue line). Our objective was to identify the subset of important edges that preserved the bias within 20% of the original value. Since each model contains hundreds of thousands of edges, it was computationally infeasible to consider all of them; therefore, we limited the search to the top N (N ranging from 600 to 3000) edges and selected the k value corresponding to the desired bias threshold. We described Figure 7 in detail in Section E in appendix.
>
> 2-1 **EAP explanation** The first part of Equation 1 can be expanded using Taylors series as follows.
>
> $L(x_{\text{clean}} \mid \text{do}(E = e_{\text{corr}}))  \approx  L(x_{\text{clean}}) + (e_{\text{corr}} - e_{\text{clean}})^{\top}
> \frac{\partial}{\partial e_{\text{clean}}}
> L(x_{\text{clean}} \mid \text{do}(E = e_{\text{clean}}))$
>
> So Equation 1 is eventually equals to $(e_{\text{corr}} - e_{\text{clean}})^{\top}
> \frac{\partial}{\partial e_{\text{clean}}}
> L(x_{\text{clean}} \mid \text{do}(E = e_{\text{clean}}))$ and  it is clear from  the above expression that it requires two forward pass and one backward pass to compute it.
>
> 2-2 **Details on Gradient Computation for EAP:** We would like to clarify that in our framework, gradients from the external sentiment model (e.g., DistilBERT) are not backpropagated through the LLM. The external model is used solely to categorize the next tokens to compute  the  metric L. To attribute importance to internal edges, we apply Edge Attribution Patching (EAP), which perturbs LLM activations with corrupted samples and measures the resulting change in L. The metric L is computed based on the output computed based on the internal weights of the model. Hence the differentiability of the model output is required.
>
> 3-1 **Comparison with Other Debiasing Methods:**  We have now included two recent circuit editing based debiasing approach [1] and [2] for comparison in Table 4. The main takeaway from Table 4 is that, although other debiasing methods achieve a greater reduction in bias, they also lead to a larger decrease in performance on other tasks (i.e., NER and CoNLL).
>
> 3-2 **Syed et al Reference:** We have now consistently used the more recent citation, Syed et al. (2024), throughout the entire paper.
>
> Our response to your **major concern** is as follows
>
> **only Two Bias Types and Two Architectures:**  To address this concern we have added two new architectures (Qwen and Gemma) into our work.
>
> [1] Identifying and adapting transformer-components responsible for gender bias in an English language model, Chintam et al. BlackboxNLP@ACL 2023
>
> [2] BiasEdit: Debiasing Stereotyped Language Models via Model Editing, Xu et a., TrustNLP@ACL  2025

---

> ### Author Response · Authors · 2025-10-12
>
> Due to space limitations we continue our response here.
>
> **Lack of explanation for Table 2 and Figure 3** Table 2 presents the effectiveness of different corruption techniques and metrics on bias related component identification. We first compute the metric value (e.g., L1 or L2) for the original model $M_{orig}$ without any
> intervention. Next, we evaluate a model restricted to only the top-k components identified by a given metric–corruption pair ($L_1,C_1$)($L_1,C_2$)($L_2,C_1$), $L_2,C_2$ and compute the corresponding metric value.
> The change is then measured as $(M_{orig}- M_{L,C})/M_{orig}$. A smaller change indicates that the top-k components identified by that method more accurately capture the bias-related components. Accordingly, for each model and dataset, we highlight in bold the method that yields the lowest change. Lower is the change more faithful are the components identified by the corresponding configuration.
>
> In EAP, all edges are ranked by their importance for bias computation. Figure 3 reports the drop in the EAP metric value after corrupting the top x\% edges from this sorted list. The right axis shows the percentage change relative to the original model. We observe that corrupting fewer than 40\% of the edges already reduces the metric value to around 87\% of its original level. This supports our claim that bias-related components are localized, and also provides guidance on selecting an appropriate top-k threshold for comparison. For all models—except GPT-2 Large on gender bias—the drop is substantially steeper within the first 50\% of edges than beyond this point.
>
>
> **Similarity with Prior Work [1,2]:**  In [1], there was no causal analysis to find the reason of bias shown in output.
> And in 2 the focus was only on attention heads, or layers as a whole.  whereas we focused on all possible components in a much more granular level. In top components we have a mixture of heads, logit nodes, mlps and everything.  There is no reason why particularly only attention heads will be responsible for bias. None of these existing works looked at two different types of bias across different models.  We have also added [1,2] in the related work section.
>
> Please let us know if you need any further clarification on any one of your comments.

---

### Review · Reviewer_tbkQ · 2025-09-17

**Summary Of Contributions:**

- uses Edge Attribution Patchingto localize edges in LLM, gpt2 small/large and llama2, that carry demographic and gender bias.
- proposes two novel (general-context) semantic bias metrics.
- studies the effect of variations of corruption techniques.
- demonstrates localization of bias to a relatively small set of layers.
- analyzes the stability of important edges with respect to w.r.t. 1. Grammatical Structures, 2. Different Types of Bias, 3. Finetuning
- leverages identified important edges to debias LLMs.

**Additional Comments:**

Metrics:
- I assume the temperature is 1 throughout the experiment. Is that correct? It would be great to mention it in the appendix A.
- Both metrics proposed rely on externally defined positive/male and negative/female sets. I’m especially concerned about the sentiment proxy using distilled-bert. As demographic-bias conclusions depends on the truthfulness/validity of the sentiment proxy, please justify this choice.
- In appendix A, you claimed that "beyond 10 tokens the amount of bias present in the model doesn’t change that much". I wonder if you have can provide a simple plot showing the effect of k when $k \in [1,10]$ and  $k \in \{20, 50, 100\}$ to illustrate the effect of $k$.
- In table 2, you showed that $(L_2,C_2)$ leads to the smallest changes in metric, so you choose them for the future experiment, since "smaller change implies that set of important edges identified by EAP performs almost similarly as the whole model where all the edges have been considered". I wonder if the two metrics inherently have different sensitivity. Intuitively $L_1$ is more sensitive than $L_2$, as $L_1$ is two-sided, so choosing the smallest raw changes in metric might lead to choosing the least sensitive metric instead. Hence, comparing the normalized changes would make more sense to me, where the normalization takes the sensitivities into account. I hope I make my point clear. In fact, I also observed that the L1's changes are significantly larger that that of L2, which supports my concerns.

Finetuning experiment:
- it would be good to provide the training recipes in the appendix.
- why these two datasets? Could you provide a brief justification? I would imagine a suite like glue or similar might be a better choice?

Debiasing:
I do not quite understand the arguments on page 9 (and figure 3/4.) In particular, is the number the relative percentage changes compared to the pretrained model? Is it the case that figure 3 shows your debiasing approach and that figure 4 shows Drechsel and Herbold (2025) approach? Is the model you presented in figure 4 llama2 or llama3.2 (I assumed it's a typo)? Is the eval recipes the same throughout all the evaluations, e.g. sampling strategy + hyperparams, prompts? Why is the number in $\delta$ bias differ so significantly, gpt2-large DSS2 v.s. the others? Probably it would be good to also provide the actual numbers if those percentages are relative.

Others:
It is very interesting that for demographic bias under grammatical variation, the sets of important edges exhibit minimal overlap, while there exists a substantial degree of overlap for gender bias. It would be great if you could expand discussion and hypothesize mechanisms a bit more. Also, it feels worthy to promote a compact version of Appendix F (SAEs) to the main text to triangulate your causal story and compare against an alternative approach.

**Audience:**

Yes

**Audience Explanation:**

Researchers in mechanistic interpretability, AI safety, fairness/privacy, and the broader ML community will find this problem interesting, as this is a trending topic nowadays. The paper provides concrete evidence with the lens of MI that (i) bias concentrates in relatively few layers, (ii) they also affect LM's general capabilities, and show (iii) various stability results. These are useful starting points for both empirical MI and fairness work.

**Claims And Evidence:**

Yes

**Claims Explanation:**

The experiments are reasonably detailed given the computational resources. The methodology is intuitive and results generally support the central claims.

**Requested Changes:**

- in the second last paragraph on page 7, "... edges in $DSS1_{pos} (GSS1_{pos})$ and $DSS1_{neg} $ $GSS1_{neg}$, in Figure 2..." should be "$DSS1_{pos} (GSS1_{pos})$ and $DSS1_{neg} (GSS1_{neg})$" right?
- in the first paragraph of section 3, "An edge refers generally represents..."
- in table 4, the column name is Debiased Llama2 but the caption is llama3.2
- the writings, in particular the clarity, should be substantially improved

---

> ### Author Response · Authors · 2025-10-12
>
> Thank you for your detailed comments. We have considered all your suggestions and submitted a revised draft. Our response to your **Requested Changes** is as follows.
>
> - **$DSS1_{pos}(GSS1_{pos})$and $DSS1_{neg}$, Confusion:** We have fixed the line in the  second last paragraph on page 7 as $DSS1_{pos}(GSS1_{pos})$  and  $DSS1_{neg}(GSS1_{neg})$.
>
> - **"An edge refers generally...:**  We have fixed this line as "An edge generally represents."
>
> - **Table 4, the column name and Caption Name** We have fixed the column name and the caption in Table 4. The model name should be llama3.2.
>
> - **Clarity in Writings:** We have fixed the writing wherever you have mentioned in your comments. Please let us know if there is any other place where you think that the clarity should be improved.
>
> Our response to your **Additional Comments** is as follows:
>
> - **Temperature:** Yes, The temperature is set to 1 for the experiments. We have included this details in Section A in Appendix.
>
> - **Relying on distilled-bert:** We particularly used \url{https://huggingface.co/distilbert/distilbert-base-uncased-finetuned-sst-2-english} which is finetuned on sst2 dataset which is a widely used sentiment classification dataset and also gives $91\%$ performance on this widely used data. There exists a number of sentiment classification task [1,2,3] where distillbert has shown good performance in sentiment classification.
>
>   [1]  Online News Sentiment Classification Using DistilBERT
>
>    [2] Utilizing DistilBERT Transformer Model for Sentiment Classification of COVID-19's Persian Open-Text Responses — Masoumi \& Bahrani (2022)
>
>    [3]  Analyzing the Generalizability of Deep Contextualized Language Representations for Text Classification — Berfu Buyukoz (2023)
>
> - **Bias Beyond 10 Tokens:** In Figure 6 in Appendix we have shown the effect of increasing the number of tokens on bias. It can be observed from Figure 6, that there is not much variation in bias value of the model beyond top 10 tokens.
>
> - **Sensitivity of Metrics in Table 2:** We have now normalized the values in Table 2. For each value it is showing the relative change with respect to the original value of the metric in the model.
>
> >Finetuning Comments
>
> - **Finetuning Details:** In Section G in Appendix we hae provided all the details related to finetuning setup.
>
>  - **Justification for the Two Datasets:** There has been a substantial body of work [4,5] in NLP demonstrating that targeted fine-tuning can effectively help in debiasing language models. Building on this line of research, we adopted the first category of datasets in which we provided debiased counterparts for all biased outputs generated by the pretrained model.
>
>     For the second category of datasets, the objective was to use data that had no overlap with the biased content. In this case,   using the Shakespeare dataset (or any similar corpus) would have served the same purpose. The Shakespeare dataset is a widely used and versatile small corpus in Natural Language Processing, particularly for language modeling and text generation tasks. Hence, we opted to use the Shakespeare dataset for this part of our study.
>
>     [4] Empowering Fine-tuning for Debiasing Pre-trained Language Models Ghanbarzadeh et al. (2023)
>
>     [5] FineDeb: A Debiasing Framework for Language Models, Hedge et al (2023)
>
> - **Figure 3/4 Explanation:** The numbers presented in Table 4 are relative decrease or increase with respect to the values of the original model. In table 5 in Appendix, we have showed the bias values of the original model. For debiasing we have used exactly similar prompts, hyperparameters and sampling strategy. We are not aware of any pretrained debiased Llama2 model. Hence for the pretrained model in Debiased LLama we have used Llama3.2 model. However rest of the debiasing experiments in Llama is done using only Llama2.
>
> Please let us know if you need any further clarification regarding anyone of your comments.

---

### Review · Reviewer_DiFn · 2025-10-14

**Summary Of Contributions:**

The authors discuss the important problem of understanding bias in LLMs, approaching it from the perspective of identifying the specific model edges responsible for such bias. To this end, they define metrics to quantify bias and study how these metrics change as different edges in the model are ablated.


## Questions
1. The authors refer to relevant literature, but one of the most notable papers on dissecting bias in neural networks is the work by Bolukbasi et al. The paper is mentioned in the context of a particular dataset and not the methods (which in many ways share the essence of Mech Interpretability work of using linear probes/SAEs)
2. Why do the authors choose GPT-2 Llama? The model family and size of course don't matter to the question but was intrigued by the choice since they are quite old and more powerful models of similar or smaller sizes are available.
3. Where do the contexts come from for the Demographic datasets?
4. I think the authors could clarify their exposition in several places (especailly their results), some of which I mention below"
- How were the neutral nationalities—such as “Emirati” and “January to return broadcaster”—identified? Please elaborate on this.
- How are the edges determined? How many forward passes were performed? It seems that one would need as many forward passes as there are total edges in the model. How is this made computationally efficient? Conversely, please elaborate on how attribute patching is implemented. A clear description would greatly enhance the reader’s understanding of both the methodology and its
- There are incorrect uses of parentheses that obscure meaning and grammar. For example, on page 6, the statement that “the sum of the probabilities of next-order predictions is greater than or equal to the sum of the probabilities of next-order predictions” reads as a tautology. Perhaps the parentheses should be removed? On a related note, why not generate k sentences and classify them as positive or negative, rather than taking the original sentence? Wouldn't that make a more biased (which I guess is the desire here) dataset?


## Concerns
I think the above can be easily fixed. I am more concerned about the broader methodology:

The first major point relates to the stability of the findings. It appears that the identified bias edges are unstable: as the authors note, fine-tuning leads to a different set of edges than those originally found. Moreover, the edges vary across different tasks—for example, demographic bias edges differ from those associated with gender bias, and the same holds for grammatical reformulations.

Given the above, I find the claim on page 7—that “these results suggest that the circuits for bias demonstrate similar generalizability patterns across both small and large models”—to be quite misleading.

The authors also find that attempts to mitigate bias lead to a decrease in performance on other natural language tasks. Further suggesting the lack of a clear set of edges and potential confounders. On a related note, what does \delta in Table 3 represent, and how is it defined?

Given these findings, it almost feels like p-hacking—though I am sure that this is certainly not the authors’ intention. In a sufficiently large network, one can almost always identify some subset of edges that appear more or less important for a given task!

**Audience:**

Yes

**Audience Explanation:**

I think the question phrased by the authors is very interesting and of interest to TMLR.

**Claims And Evidence:**

No

**Claims Explanation:**

See above.

**Requested Changes:**

See above.

---

> ### Author Response · Authors · 2025-10-14
>
> Thank you for you detailed comments. We have considered all your suggestions and submitted a revised draft. Our response to your concerns is as follows.
>
> **Stability of Findings** We would like to clarify that the circuits identified across grammatical variations and different types of bias and finetuning are distinct, this is one of the main conclusions of our work. To the best of our knowledge, this is the first study to perform such an analysis. On page 7, when we referred to a “similar generalisability pattern,” we meant that the relationship between circuits for demographic and gender bias in GPT-2 (i.e., low similarity) is consistent with what we observe in LLaMA-2. Apologies for the confusion. We have rephrased the sentence as: “The similarity and dissimilarity patterns across different circuit variations are consistent across models".
>
> **Definition of delta bias** Delta bias shows percentage of change in the bias value of the model after the debiasing approach has been applied. Mathematically the formula is defined as follws $\frac{|bias_{orig}-bias_{modelediting}|}{bias_{orig}}*100$. The original bias values are reported in Table 5 in Appendix.
>
> **Clarification of Bias Claim** Our claim regarding the proposed debiasing approach is that it effectively reduces model bias while causing substantially less degradation in performance on other NLP tasks compared to existing methods. This is happening since a number of tasks can share circuits. There exists a number of works [3,4] in existing literature which shows that different tasks share circuits in LLMs. Table 4 presents a comparison with other debiasing techniques. As shown, although some existing methods achieve a larger reduction in bias, they also lead to a significantly greater drop in performance on other tasks. This demonstrates that our approach can mitigate bias with minimal perturbation to the model’s general performance. Such trade-offs between debiasing and task performance have also been reported in prior literature [2].  The fact that EAP approach can identify components that are causally relevant to bias, is shown in Figure 7 in Appendix where we show that using only top-k edges identified from EAP we could achieve 80% of the bias shown by the original model. That's why our proposed debiasing method introduces far less perturbation to other tasks.   Another key advantage of our debiasing method is that it does not require retraining the model.
>
> Please let us know that if there is any other comment which you would like us to clarify.
>
> [2] "Identifying and adapting transformer-components responsible for gender bias in an English language model"-  2023
>
> [3] "Circuit Component Reuse Across Tasks in Transformer Language Models"-ICLR 2023
>
> [4] "Towards Interpretable Sequence Continuation: Analyzing Shared Circuits in Large Language Models"-EMNLP 2024

---

> ### Author Response · Authors · 2025-10-14
>
> Our response to your questions is as follows.
>
> 1. **Bolukbasi Debiasing Work**  We specifically focused on studies that applied mechanistic interpretability techniques to analyze bias, which is why this work was not initially included. However, thank you for pointing it out, we have now added it to the updated related work section.
>
> 2. **Only GPT2 and Llama2** We have now added Qwen 20.5b and Gemma2-2b model in the updated pdf. If you have any other particular model in mind please let us know.
>
> 3. **Demographic Dataset Context** We have chosen the context of demographic bias from the existing work on demographic bias from [1]
>
>      [1] "Nationality Bias in Text Generation"-EACL 2023
>
> > Clarification of Results
>
> - **Neutral Nationality Selection Process** Out of all the nationalities mentioned in the work [1], the choice of neutral terms was guided by examining the model’s output distributions: we selected terms for which the predicted probabilities of next positive and negative tokens were approximately equal, ensuring that they did not introduce unintended bias.
>
> -  **Determination of Edges:**  EAP methodology gives the importance of all the edges. Once that is done, then we do an  ablation study demonstrated in Figure 7. Figure 7 illustrates the effect of retaining only the top-k edges on the model’s bias. From this figure, we determined the value of k for which the model’s bias remained within 20\% of the bias observed in the original model (indicated by the blue line). Our objective was to identify the subset of important edges that preserved the bias within 20% of the original value. Since each model contains hundreds of thousands of edges, it was computationally infeasible to consider all of them; therefore, we limited the search to the top N (N ranging from 400 to 3000) edges and selected the k value corresponding to the desired bias threshold. We described Figure 7 in detail in Section E in appendix.
>
>  -  **Computational Efficiency of EAP** EAP computation for each pair of clean and corrupted sentences require only two forward pass and one backward pass.  The importance of each edge in EAP is determined by the following expression.
> $(e_{\text{corr}} - e_{\text{clean}})^{\top}  \frac{\partial}{\partial e_{\text{clean}}}  L(x_{\text{clean}} \mid \text{do}(E = e_{\text{clean}}))$
>
>  In the above expression $e_{\text{corr}}$ and $e_{\text{clean}}$ can be computed from the forward pass and the other part can  be computed from the backward pass. More details about the implementation is described in the paper [2].
>
> [2] "Attribution Patching Outperforms Automated Circuit Discovery"- Syed et al 2024
>
> -  **Generating K sentences** We take the top $k$ tokens and we form $k$ sentences where the initial context comes from the input prompt in demographic context and the next token is replaced by kth token. These sentences are classified as positive and negative sentences.

---

> ### Comment · Reviewer_DiFn · 2025-10-16
> **Quick update**
>
> Thank you for the response, while I digest what you posted, could you please address
>
> > Given these findings, it almost feels like p-hacking—though I am sure that this is certainly not the authors’ intention. In a sufficiently large network, one can almost always identify some subset of edges that appear more or less important for a given task!
>
> Note that this same critique would not have applied if different rephrasings and/or topics would have found the same set of edges (since the probability of that happening by chance decreases at least as fast as the model size)

---

> > ### Author Response · Authors · 2025-10-16
> >
> > Thank you for your response.
> >
> > To address your concern, the first point we would like to clarify is that the proposed approach successfully identifies age-related bias. As shown in Figure 7, our experiment demonstrates that the selected edges can reproduce approximately 80% of the original bias.
> >
> > Secondly, it is likely that some of the edges identified by our approach overlap with those that are important for the CoLA and NER tasks. This explains the slight decrease in performance observed in these tasks. Similar behaviour has also been reported in existing debiasing work by model editing [2], where debiasing often leads to some degradation in performance on unrelated tasks. As you also noted, "since the probability of such overlap decreases with model size", similar observation can  be found in Table 4, where the performance impact on other tasks is much smaller in Qwen, Gemma, and LLaMA2 models compared to GPT-2 Large.
> >
> > Please let us know if there is anything else we can clarify.

---

> > > ### Comment · Reviewer_DiFn · 2025-10-16
> > >
> > > FYI The probability I was referring to is for different rephrasings and/or topics would have found the *same* set of edges

---

> > > > ### Author Response · Authors · 2025-10-16
> > > >
> > > > Thank you for engaging with our response.
> > > >
> > > > Figure 4 shows that different rephrasings activate different sets of edges; however, the overlap is not entirely zero. This demonstrates that bias circuits depend on grammatical variations. The causal relevance of the edges used in Figure 4 can be observed in Figure 7, where we show that using only the top-k edges of a model successfully reproduces the bias in the corresponding grammatical variations.
> > > >
> > > > Additionally, as shown in Table 4, we initially considered two datasets representing different bias directions within the same phrasing — positive (male) and negative (female). It can be observed that, across all models, the overlap between the circuits corresponding to these two variations is relatively high. This suggests that the same set of components are responsible for both male and female (or positive and negative) bias within a given phrasing.  Figure 8 in Appendix which uses another causal attribution approach also gave same conclusion. These findings confirm that the overlap and non-overlap patterns observed in Figure 4 are not accidental.
> > > >
> > > > Please let us know if you have any further concerns.

---

> > > > > ### Author Response · Authors · 2025-10-17
> > > > >
> > > > > To further strengthen our findings related to Figure 4, we have added three additional experiments in the updated version of the paper.
> > > > >
> > > > > **First**, for the SAE-based approach (another causal attribution method), Figure 8 shows a similar overlap pattern to that in Figure 4, indicating that multiple causal attribution techniques produce consistent analyses.
> > > > >
> > > > > **Second**, Table 8 reports that the important edges identified by the EAP method have a 75% overlap with those found by the SAE approach. This further supports our argument that the edges discovered by EAP are not random.
> > > > >
> > > > > **Finally**, in Figure 7, we compare the results of the EAP approach with those obtained using randomly selected edges. As shown, across all cases, the randomly chosen edges consistently perform worse than the edges identified by the EAP method
> > > > >
> > > > > We believe that the above justifications provide strong evidence supporting the faithfulness of the edges identified through the EAP approach.

---

### Author Response · Authors · 2025-10-17

We have updated the revision by adding three additional experiments.

**First**, for the SAE-based approach (another causal attribution method), Figure 8 in Appendix shows a similar overlap pattern to that in Figure 4, indicating that multiple causal attribution techniques produce consistent analyses.

**Second**, Table 8 in Appendix reports that the important edges identified by the EAP method have a 75% overlap with those found by the SAE approach. This further supports our argument that the edges discovered by EAP are not random.

**Third**, in Figure 7, we compare the results of the EAP approach with those obtained using randomly selected edges. As shown, across all cases, the randomly chosen edges consistently perform worse than the edges identified by the EAP method

We believe that the above justifications provide strong evidence supporting the faithfulness of the edges identified through the EAP approach.

---

### Author Response · Authors · 2025-10-26

Dear Reviewers,

Please let us know if you need any further clarification regarding your comments.

Thanks
Authors of Submission 5727

---

### Author Response · Authors · 2025-11-18
**Correction Regarding the Visibility of Responses**

Dear Reviewer 1xiq and tbkQ,

Unfortunately, we just realized that our responses posted on 12th October may not have been visible to you, as we mistakenly didn’t set their visibility to ‘everyone.’ This has now been corrected. We hope you will consider our responses when making your final decision.

Thanks
Authors

---

### Decision · Action_Editor_dt2o · 2025-12-15

**Recommendation:** Accept as is

**Additional Comments:**

This paper used mechanistic interpretability approaches to study social, demographic, and gender biases of LLMs. After rebuttal all reviewers agreed it met the criteria for publication at TMLR. Concerns remained around novelty and impact, but those are not criteria for TMLR publications.

**Audience:**

Yes

**Audience Explanation:**

Review unanimously agreed.

**Claims And Evidence:**

Yes

**Claims Explanation:**

Review unanimously agreed.